# Three Frequency Up-Converting Piezoelectric Energy Harvesters Caused by Internal Resonance Mechanism: A Narrative Review

**DOI:** 10.3390/mi13020210

**Published:** 2022-01-28

**Authors:** Jian Liu, Yongling Lu, Zhen Wang, Sen Li, Yipeng Wu

**Affiliations:** 1Research Institute of State Grid Jiangsu Electric Power Co., Ltd., Nanjing 211103, China; liujiandky@163.com (J.L.); 15105182955@163.com (Y.L.); wangzhenscut@163.com (Z.W.); 2State Key Laboratory of Mechanics and Control of Mechanical Structures, Nanjing University of Aeronautics and Astronautics, Nanjing 210016, China; lisensen@nuaa.edu.cn

**Keywords:** vibration energy harvesting, frequency up-conversion, internal resonance, low frequency, piezoelectric

## Abstract

Low frequency mechanical vibrations are ubiquitous in practical environments, and how to efficiently harvest them with piezoelectric materials remains a challenge. Frequency up-conversion strategies—up-converting low frequency vibrations to high frequency self-oscillations—can improve the power density of piezoelectric materials. This paper mainly introduces a kind of frequency which up-converts piezoelectric energy harvesters based on an internal resonance mechanism, compared with the other mechanisms caused by mechanical impact, mechanical plucking, etc.; the internal resonance-based harvesters can up-convert the frequency under a condition of lower excitation level, less energy loss, and less wideband operation bandwidth. Benefits to practical vibrations also exist in these multi-degree-of-freedom nonlinear dynamic systems. Moreover, the value of the frequency up-conversion factor based on the 1:2:6 internal resonance mechanism can reach as much as six so far, which is also a quite a high frequency up-conversion value.

## 1. Introduction

As an important part of the network topology of Internet of Things, wireless sensor networks which can directly sensor the physical world and produce endless big data are becoming increasingly widespread [1]. However, most wireless sensor nodes are still powered by batteries at present; the maintenance expenses for battery charging or replacement are unimaginable, provided that the number of wireless sensor systems is large. Moreover, a large number of waste batteries will also put tremendous pressure on the environment [2]. Scavenging ambient energies related to these autonomous devices, for instance, solar [3], wind [4,5], temperature gradient [6], radio frequency [7], vibrations [8], and converting them into electrical energy, have become a good solution to the problems mentioned above. This paper mainly focuses on vibration energy harvesting, since mechanical vibrations are ubiquitous in our practical environment [9]. Vibration energy harvesting technology can be roughly divided into four types: electromagnetic [10], piezoelectric [11], electrostatic [12], and triboelectric [13]. It should be noted that the classic electromagnetic induction technology was adopted earliest [14]; research on electrostatic and triboelectric energy harvesters is becoming more and more popular [15]. The piezoelectric transducer has the advantages of easy integration, constant coupling performance, etc., but it does have the disadvantages of large output impedance and dielectric loss under low frequency conditions [16]. For a typical piezoelectric vibration energy harvesting system, it mainly includes a piezoelectric oscillating structure, an electrical energy extraction interface circuit, and a power management unit. The piezoelectric oscillator directly converts mechanical vibration energy into electrical energy, and determines the potential maximum power output of the harvesting system. The interface circuit efficiently extracts the electrical charge from the piezoelectric transducer, and determines the practical output power of the harvesting system. The power management unit has functions of an electrical energy storage and voltage regulator, and balances the various harvesting power with the consuming power in the autonomous energy system. This review paper mainly talks about piezoelectric oscillating structures, since they are the sources of the transferred electrical energy.

## 2. Piezoelectric Oscillating Structures

### 2.1. Resonant Frequency Matching Strategy

A cantilevered beam bonded with piezoelectric elements is the most common piezoelectric oscillating structure. This kind of structure is able to amplify the ambient vibration, provided that the first-order natural frequency matches the peak of the excitation vibration spectrum [17]. However, the output electrical power will drop drastically if the vibration frequency deviates from the natural frequency. With real-word applications where vibration energy is likely to be broadband or varying, the ideal linear cantilevered beam structures are still limited [18]. A resonant frequency matching strategy is therefore proposed and considered to be a classical approach to addressing the challenges of inadequate power density and narrow operation bandwidth. Many resonant frequency tuning [19], multimodal [20], and nonlinear wideband [21] oscillating structures have been developed over the past decade.

The piezoelectric oscillating structures with resonant frequency tuning mechanisms are usually realized by changing their equivalent stiffnesses or mass. Sun et al. developed a resonant frequency tuning harvester based on magnetic force, as shown in Figure 1 [22]. The oscillating structure could adjust the horizontal and vertical magnetic distances synchronously, which tunes the resonant frequency in a maximal range of 51 Hz to 110 Hz practically. Tang and Yang also proposed a resonant frequency tuning harvester based on a couple of magnetic forces, as shown in Figure 2 [23]; the harvester consisted of two cantilevered beams which corresponded to two degrees of freedom (DOFs); the equivalent stiffness modulation caused by the nonlinear magnetic force broadened the operation bandwidth effectively. The experimental results showed that the resonant frequency of the proposed structure varied in the range of 23 Hz to 30 Hz. Besides the above frequency tuning mechanism, Rui et al. proposed a piezoelectric oscillating structure in which the resonant frequency could be tuned passively through a variable centrifugal force. This harvester was suitable for rotational mechanical energy and had the characteristic of wideband behavior [24]. Wang et al. also proposed a piezoelectric energy harvesting structure which can self-adjust the resonant frequency through changing the effective length of the cantilever. The average output power of the designed prototype could achieve as much as 107.4 μW under wheel rotation speeds ranging from 177 to 796 rpm [25]. Besides, it could be selected as a disturbing torque, absorbing in rotating environments [26].

The multimodal piezoelectric oscillating structures usually have multiple resonant frequencies through multiple DOFs in operation bandwidths. For example, Dechant et al. integrated several piezoelectric cantilever beams with one basic frame. It is capable of resonating at various frequencies by properly selecting the length and tip mass of each beam, and thus provides high voltage over a wide frequency range [27]. Figure 3 shows another kind of piezoelectric harvesters with multiple DOFs. The structure is like a spiral rectangular cross-section piezoelectric cantilever, and its first three modes can efficiently convert the external excitation into the expansion and torsion strain energy of the harvester [28]. In addition, the cantilevered structure proposed by Dechant et al. also has high-order modes, but the resonant values over the second-order modes are not in the range of practical operation frequency. Therefore, the simple cantilevered structure is usually considered as a single mode oscillator in the energy harvesting research field.

Nonlinear piezoelectric oscillators targeted on the enhancement of the operation bandwidth are another mechanism to match the practical excitation spectrum [29]. Among them, monostable, bistable, and multi-stable oscillators which are classified according to the number of their stable positions are widely researched [30]. Monostable oscillators generally achieve the broadband responses through two methods: Duffing-type or piecewise-linear mechanisms. The Duffing-type oscillator denotes that the nonlinear stiffness of the system changes smoothly, while the piecewise-linear oscillator refers that the equivalent stiffness jumps during the vibration, such a mechanism can be physically realized by means of adding mechanical stoppers to conventional linear oscillators [31]. For a nonlinear bistable dynamic system, the potential energy function has two potential wells separated by a potential barrier. Depending on the excitation level, the bistable system may exhibit three distinct trajectories, which are intrawell oscillation, chaotic interwell vibration, and interwell oscillation [32]. Bistable oscillators can be realized by magnetic force [33], purely elastic bucking approach [34], and recompressed spring [35], etc. For instance, Andò et al. developed a bistable oscillator by using a simple bucking beam, as shown in Figure 4. When the beam was excited to switch between the two stable states, the attached mass would impact the piezoelectric elements symmetrically arranged on both sides of the harvesting device [34]. Generally speaking, excellent energy harvesting performance happens when the nonlinear dynamic system exhibits chaotic interwell or interwell oscillations; a quite high excitation level is then necessary to surmount the potential barrier. This threshold excitation amplitude can be decreased by designing multi-stable oscillators. Zou et al. introduced this kind of piezoelectric oscillators. The results demonstrated that the multi-stable piezoelectric harvester also had a relatively high output power and wide operation bandwidth [36]. Finally, Table 1 summarizes the key features of the resonant frequency matching oscillators mentioned above.

### 2.2. Frequency Up-Conversion Strategy

Recently, an increasing number of researchers have realized that most environmental vibration sources are low frequencies or ultra-low frequencies, e.g., wave heave motions [37], wind-induced vibrations [38], and biological motions [39]. In these cases, a resonant frequency matching strategy would increase the complexity of structure designation and optimization: (1) weak spring stiffness and heavy proof mass which leads to large dimensions and flimsy oscillating structure; (2) lower power density but larger dielectric loss of piezoelectric elements; (3) the response amplitudes are usually large, leading to higher requirements of the structure fatigue life and the integration of the piezoelectric ceramic, and the depolarization phenomenon of piezoelectric ceramics could also easily happen; (4) the electrical energy extraction efficiency is difficult to improve.

Another lies in the fact that the output power of the piezoelectric harvester can be simply considered as proportional to the cube of its oscillating frequency [40]; harvesters with low resonant frequencies usually suffer from deteriorated electrical power generation. Consequently, frequency up-conversion (FUC) strategy, up-converting low frequency excitations to high frequency self-oscillations, and probably first proposed by Kulah and Najafi in 2008, may mitigate the above issue [41]. This strategy bridges the gap between the high-frequency response and low-frequency input physically, through several mechanisms: mechanical impact [42], mechanical plucking [43], impulse-like acceleration [44], impulse-like magnetic force [45], and internal resonance [46], to name a few.

The mechanical impact mechanism is the most classical method to achieve the FUC effect. Huang et al. have proposed a mechanical impact FUC oscillator based on two asymmetric cantilevered beams [47], as shown in Figure 5. The stainless-steel beam with low natural frequency value could easily match the external low frequency excitation and drastically vibrate. As soon as the stainless-steel beam impacted the silicon cantilevered beam, the mechanical energy existed in the harvester system would be directly transferred into the silicon beam through elastic collision; self-oscillation at its natural frequency would immediately happen. Because the natural frequency value of the silicon beam was larger than that of the stainless-steel beam, the FUC effect was therefore realized. The silicon beam should oscillate continuously due to the periodic impacts. The electromechanical conversion was obtained by means of the positive piezoelectric effect. Experimental results demonstrated that this proposed FUC mechanism could up-convert the excitation frequency from 40 Hz to 1012 Hz. Besides, Yang et al. [48] and Chen et al. [49] proposed a type of vibration energy harvesters based on a piezoelectric cantilever and magnets on both of its sides. The magnetic attraction would amplify the impact energy by accelerating the approaching velocity between the beam and the magnet. Therefore, high-frequency oscillation occurs in the piezoelectric beam when its motion is suddenly stopped by the magnets. The harvesting power and energy of the structure proposed by Yang et al. could achieve 20 mW and 190 μJ at an optimized resistive load of 30 kΩ.

Pozzi and Zhu have proposed a classical mechanical plucking FUC structure based on picks and piezoelectric cantilevers with high resonant frequencies [50]. The picks moved with the external ultra-low frequency excitation; as soon as they plucked the piezoelectric cantilevers, part of the mechanical motion energy of the harvester system is transferred into the elastic potential energy of the cantilevers, and self-oscillations at their natural frequencies would occur passively. From a physical point of view, the principle of mechanical impact is like playing a piano, while the principle of plucking excitation is equivalent to playing a guitar. However, the FUC process would only make noises instead of music, and the energy loss for converting frequency is inevitable. Periodic collision and large deformation also require a relative high fatigue life standard of the structures.

Impulse-like magnetic force is a noncontact mechanism for achieving the FUC effect; the magnetic plucking force between the two movement parts is gentler, and the energy loss during the frequency conversion is much less compared to the mechanical plucking approach. Tang et al. proposed this kind of FUC oscillator based on a couple of permanent magnets and two-stage vibratory structure [51], as shown in Figure 6. The first stage picked up ambient low frequency vibration and excited the second stage to vibrate at its resonant frequency, thereby realizing FUC by magnetic plucking force and improving power generation capability. The experimental results demonstrated that this FUC harvester can effectively collect the environmental vibration energy below 1 Hz. Similar to the mechanical impact and plucking approaches, these FUC mechanisms can obtain a quite large factor value (probably larger than 10) of frequency up-conversion, but a high excitation level is also necessary. Otherwise, it will be difficult to achieve the FUC effect.

The impulse-like acceleration approach is another noncontact mechanism which can be selected to realize the FUC effect. Han and Yun proposed this kind of FUC harvester based on a buckled bridge beam that snapped through between two stable states, inducing shock acceleration on the attached high frequency piezoelectric resonators. In order to optimize the external acceleration threshold, they tested various flexible sidewall materials and modified its operation mechanism; the finial value of the threshold acceleration value was 0.5 g [44].

Because the FUC effect can be touted as the break-through to boost the functionality of low frequency vibration energy harvesters, this strategy is attracting great research interest, and showing the highest efficiency to date; another FUC approach based on internal resonance mechanism was proposed in references [52,53,54]. The internal resonance phenomena especially exist in multi-DOF nonlinear dynamic systems, provided that the resonant frequencies are carefully designed. When the internal resonance phenomenon happens, part or most of the mechanical energy can be transferred from the first DOF to the second or third DOFs. Take an example of 1:3 internal resonance-based FUC harvester; if the resonant frequencies corresponding to the two DOFs *ω*_1_:*ω*_2_ equal 1:3, and the dynamic equilibrium equation of the system has cubic terms, 1:3 internal resonance will be induced. Normally, the output frequency should be up-converted three times, compared with the excitation frequency [55,56]. Finally, Table 2 compares several classical FUC oscillators especially designed for low-frequency vibration energy harvesting.

## 3. FUC Oscillators Based on an Internal Resonance Mechanism

### 3.1. 1:3 Internal Resonance System

Figure 7a depicts the 1:3 internal resonance-based FUC harvester; it consists of two magnetically coupled cantilevers, a base frame, and two magnets. It is worth noting that many studies have already proposed this type of magnetically coupled structure, but most of them were designed for broadband vibration energy harvesting. For example, Wu and Lee [57] presented a similar harvester with two DOFs, but the original ratio of resonances was 1:1.78. The nonlinear harvester had a wideband characteristic, but could not realize the FUC effect.
(1)$$\begin{array}{l} \left[\begin{array}{l} M_{1}\\ M_{2} \end{array}\right]\ddot{x}=\left[\begin{array}{l} M_{1}\;\;0\\ \;0\;\;M_{2} \end{array}\right]\left[\begin{array}{l} {\ddot{x}}_{1}\\ {\ddot{x}}_{2} \end{array}\right]+\left[\begin{array}{l} D_{1}\;\;0\\ \;0\;\;D_{2} \end{array}\right]\left[\begin{array}{l} {\dot{x}}_{1}\\ {\dot{x}}_{2} \end{array}\right]+\left[\begin{array}{l} K_{1}-a_{1}\;\;a_{2}\\ \;a_{1}\;\;K_{2}-a_{2}+\frac{\alpha^{2}\omega^{2}}{C_{0}\omega^{2}-j\frac{\omega }{R}} \end{array}\right]\left[\begin{array}{l} x_{1}\\ x_{2} \end{array}\right]\\ \;\;\;\;+\left[\begin{array}{l} \;b_{1}\;b_{2}\;-c_{1}\;c_{2}\\ -b_{1}\;b_{2}\;c_{1}\;-c_{2} \end{array}\right]\left[\begin{array}{l} x_{1}^{3}\\ x_{2}^{3}\\ x_{1}^{2}x_{2}\\ x_{1}x_{2}^{2} \end{array}\right] \end{array}$$

This is assuming that the harvester has the two DOFs *x*_1_ and *x*_2_, which are the vibration displacements of the low-frequency beam and the high-frequency beam, respectively. According to reference [52], the final dynamic equilibrium equation of the system can be expressed in Equation (1), where the parameters *M_i_*, *K_i_*, *D_i_* (*i* = 1, 2) are the proof mass, the global equivalent stiffness and the damping coefficient of the harvester system, respectively; *x* is the external excited displacement and *ω* is its angular frequency; the parameters *a_i_*, *b_i_*, *c_i_* (*i* = 1, 2) are the intermediate variables used to express the nonlinear magnetic force, for a given harvester structure of which the original magnetic distance is also fixed, they are the constant values; a simple resistor *R* is assumed to be directly connected to the piezoelectric element to evaluate the output power of the harvester; *C*_0_ is the equivalent capacitance of the piezoelectric material and *α* is its electromechanical coupling coefficient.

From the dynamic equilibrium equation, it is clearly found that the harvester has two DOFs, *x*_1_ and *x*_2_, and the equivalent restoring force has linear [*x*_1_ *x*_2_]^T^ and cubic [*x*_1_^3^ *x*_2_^3^ *x*_1_^2^*x*_2_ *x*_1_*x*_2_^2^]^T^ terms. In this case, if the resonant frequencies of the two DOFs *ω*_1_:*ω*_2_ equal (or are close to) 1:3, and the external excitation frequency matches the resonant frequency of the first DOF, 1:3 internal resonance of the cubic nonlinear system is then induced. Finally, the output frequency of the voltage signal is the same as the vibration frequency of the second DOF of the harvester, which is three times higher than the excitation frequency.

Based on Equation (1), a numerical simulation tool named Matlab and Simulink with an ordinary differential equation solver can be used to investigate the time-domain behaviors of the dynamic system. According to our previous research published in [46], Figure 8 illustrates the numerical simulation results of the oscillating system under a sinusoidal excitation signal, including the displacement waveforms and the corresponding frequency response curves. When the excitation frequency is close to the resonant frequency of the first DOF (17.8 Hz), the response amplitude of the first DOF is as high as 5 mm, the vibration displacement of the second DOF contains a fundamental frequency (17.8 Hz) and a triple frequency component (53.4 Hz). In this case, the resonant frequency of the second DOF affected by the nonlinear magnetic force is tuned to 53.4 Hz, which satisfies the key condition of 1:3 internal resonance. However, it is worth noting that, due to the resonant frequency tuning effect, the original natural frequencies of the two DOFs should be respectively a little larger than the values of 17.8 Hz and 53.4 Hz. In addition, the cubic nonlinear terms of the magnetic force strongly depend on the vibration displacement amplitudes, hence the FUC effect is easy to be obtained with the increasing of the excited acceleration. Such FUC effects were also validated in the experimental results [46,58].

### 3.2. 1:2 Internal Resonance System

The classical piezoelectric cantilevered oscillators are widely used in the vibration energy harvesting field. If the ambient vibration frequency is ultra-low or the vibration is multi-directional, cantilevered structure will be complicated to be manufactured. Pendulum-like structure is also a common mechanical oscillator that could be excited in any direction in the horizontal plane; the resonant frequency only depends on the pendulum length and the gravitational acceleration. Consequently, a pendulum-like structure is very suitable for multi-directional and ultra-low frequency vibration energy harvesting. However, the piezoelectric elements are difficult to be integrated with these kinds of oscillators. Figure 7b shows a schematic structure of our proposed 1:2 internal resonance-based pendulum-like oscillator [53]; it consists of a mass, a spring with bonded piezoelectric elements, and an excitation base. The spring pendulum system is based on multiple metal binder clips and assembled by pin structures in series. Such a metal clip usually has six faces to handily integrate with six piezoelectric ceramics, and successfully solves the integration problem mentioned above. As shown in the figure, though the integration with more piezoelectric ceramics can achieve a higher electromechanical coupling coefficient, the integration scheme of two pieces of piezoelectric ceramics on the bottom faces is selected, for its maximal conversion efficiency due to the uniform and large strain distributions.
(2)$$\{\begin{array}{l} M\left(L_{0}+x_{2}\right){\ddot{x}}_{1}+D_{1}\left(L_{0}+x_{2}\right){\dot{x}}_{1}+2M{\dot{x}}_{1}{\dot{x}}_{2}+Mgx_{1}=M\ddot{x}\cos x_{1}\\ M{\ddot{x}}_{2}+D_{2}{\dot{x}}_{2}+Kx_{2}-M\left(L_{0}+x_{2}\right){\dot{x}}_{1}^{2}-Mg\cos x_{1}+Mg+\frac{\alpha^{2}\omega^{2}}{C_{0}\omega^{2}-j\frac{\omega }{R}}x_{2}=M\ddot{x}\sin x_{1} \end{array}$$

This is assuming that this spring pendulum harvester has the two DOFs *x*_1_ and *x*_2_, which correspond to the angular displacement and the deformation of the spring, respectively. According to reference [53], the governing equations of motion of the system are expressed in Equation (2), where the parameters *D_i_* (*i* = 1, 2) are the corresponding damping coefficient, *M*, *K* are the equivalent mass and stiffness of the spring pendulum system, respectively; *x* is the external excited displacement and *ω* is its angular frequency; *L*_0_ is the free stretching length of the pendulum; *C*_0_ is the equivalent capacitance of the piezoelectric element and *α* is its electromechanical coupling coefficient; a simple resistor *R* is also assumed to be directly connected to the piezoelectric element and to evaluate the output power of the harvester system.

If the trigonometric functions in the governing equations of motion are transformed by Taylor series expansion, and the third and higher-order terms are omitted, the linear and quadratic nonlinear terms will appear in this 2-DOF dynamic system. According to the key condition of 1:2 internal resonance, the resonant frequency of the ‘spring-mass’ subsystem is twice as large as the swing frequency. The swinging motion induce the resonance of the ‘spring-mass’ oscillation in the second DOF, efficiently converting the mechanical energy into electrical energy through the direct piezoelectric effect.

According to the normalized parameters provided in [53], using an ordinary differential equation solver integrated in Matlab and Simulink software, Figure 9 illustrates the simulation waveforms under the sinusoidal excitation signal, including the excited acceleration, the swing angle, and the deformation of the spring structure, as well as their frequency response curves. The excited acceleration amplitude and frequency are 0.16 m/s^2^ and 1.33 Hz, respectively. The simulation results show that the ‘spring-mass’ subsystem can oscillate drastically near its resonant frequency (2.66 Hz) structure when the external excitation frequency matches the resonant frequency of the swing DOF (1.33 Hz); 1:2 internal resonance occurs in the harvester system. Besides, the results have proved that the pendulum-like oscillator could easily match ultra-low frequency excitations and obtain a relatively good energy harvesting performance.

### 3.3. 1:2:6 Internal Resonance System

Considering the FUC factor, which is defined as the ratio of the output frequency to the excitation frequency, the FUC factors of the above two internal resonance-based FUC harvesters are two and three, respectively. Since the power generation of a piezoelectric harvester can be simply calculated as proportional to the cube of the operation frequency, the small value of the FUC ratio indeed has the disadvantage.
(3)$$\left\{ \begin{array}{l} M_{1}l^{2}{\ddot{x}}_{1}+D_{1}l{\dot{x}}_{1}+M_{1}l{\ddot{x}}_{2}\sin x_{1}+M_{1}lg\sin x_{1}=M_{1}l\ddot{x}\cos x_{1}\\ M_{1}l{\ddot{x}}_{1}\sin x_{1}+D_{2}{\dot{x}}_{2}+M_{1}l{\dot{x}}_{1}^{2}\cos x_{1}+\left(M_{1}+M_{2}\right){\ddot{x}}_{2}+K_{2}x_{2}=F_{B}\\ M_{3}{\ddot{x}}_{3}+D_{3}{\dot{x}}_{3}+K_{3}x_{3}+\frac{\alpha^{2}\omega^{2}}{C_{0}\omega^{2}-j\left(\omega /R\right)}x_{3}=-F_{B} \end{array}\right.$$

To solve this issue, a 1:2:6 internal resonance-based harvester structure in which the FUC factor is six is proposed and shown in Figure 7c [54]. The harvester structure consists of a component pendulum, a pair of magnetic coupled cantilevers, and two mechanical stoppers. Normally, the three DOFs *x*_1_, *x*_2_ and *x*_3_, corresponding to the angular displacement of compound pendulum, the displacements of the low-frequency cantilever and high-frequency cantilever respectively, can provide a relevant description for this dynamic system. The governing equations of motion are then given in Equation (3), where the parameters *M_i_*, *D_i_* (*i* = 1~3), *K_i_* (*i* = 2, 3) are the proof mass, the damping coefficient, and the equivalent stiffness of the system, respectively; *x* is the external excited displacement and *ω* is its angular frequency; *C*_0_ is the equivalent capacitance of the piezoelectric element and *α* is its electro-mechanical coupling coefficient; a simple resistor *R* is assumed to be directly connected to the piezoelectric element to evaluate the output power of the harvester; *F_B_* is the coupling magnetic force between the magnets, and this magnetic force is similar to the 1:3 internal resonance-based harvester proposed in the previous subsection.

According to the theory of nonlinear dynamics, the proposed harvester structure has two coupled internal resonances, provided that the resonant frequencies of the three DOFs are configured as *ω*_1_:*ω*_2_:*ω*_3_ equals 1:2:6. When the external excitation frequency matches the resonant frequency of the first DOF, the 1:2 internal resonance phenomenon between the first and the second DOFs is induced. The low-frequency cantilever corresponding to the second DOF will oscillate drastically. Thanks to the nonlinear magnetic coupling force between the two cantilevers, the 1:3 internal resonance phenomenon between the second and the third DOFs will occur at the same time. Finally, because the piezoelectric transducer is integrated with the high-frequency cantilever corresponding to the third DOF, the piezoelectric voltage up-converts the output frequency, which is six times as high as the excitation frequency.

Still using an ordinary differential equation solver, Figure 10 shows the simulation waveforms under the sinusoidal excitation signal, including the swing angle, the vibration displacements of the two cantilevers, the piezoelectric output voltage across the optimized load resistance *R*, as well as their frequency response curves. The excited acceleration amplitude and frequency are 2 m/s^2^ and 2.03 Hz, respectively. From the frequency response curves, it is seen that the fundamental frequency of the swing signal is equal to the excitation frequency; the harmonic response (6.07 Hz, 10.13 Hz) components are due to the large swing amplitude of the pendulum. In addition, the low-frequency cantilever also vibrates drastically because of the 1:2 internal resonance phenomenon; the super harmonic components also exist due to the large vibration amplitude. The high-frequency cantilever is excited through the nonlinear magnetic coupling force, forming its power density curve; it can be found that the power spectrum contains several peaks values: probably twice, four times, six times, and eight times. It is worth noting that the graph of the power density is logarithmic; the obvious frequency components are thereby the twice (4.07 Hz) and the six times (12.17 Hz) frequencies. The value of 12.17 Hz is close to the resonant frequency of the high-frequency cantilever; it is also induced by the 1:2:6 internal resonances in the nonlinear dynamic system.

A high-speed camera is selected to capture the oscillation the proposed FUC harvester. Several key pictures with brief illustrations are given in Figure 11, from which one can see that the pendulum motion yields resonance of two beam vibrations. Once the mass of the first DOF reaches its highest position, the second DOF should be at the maximum deformation state, provided that the first DOF, together with the second DOF, are simply considered as a “spring-pendulum” model. The “spring” is at the maximum compression state. Once the pendulum mass reaches its lowest position, the “spring” should be at the maximum elongation state; the vibration displacement of the second DOF reaches its minimum. At this time, the compound pendulum has been oscillating for a quarter of its cycle, while the second DOF has been vibrating for half of its cycle. From the illustrations shown in Figure 11, it can be found that the third DOF has been oscillating for three seconds of its cycle; the FUC effect of the dynamic system is thereby directly displayed. It should be noted that Figure 11 only shows one possible oscillating mode of the multi-DOFs system, however, the other oscillating modes still yield the capability of the FUC effect and efficient low frequency vibration energy harvesting.

## 4. Conclusions

This paper mainly summarizes a type of frequency up-converting piezoelectric oscillators which can mechanically up-convert low-frequency excitations into high-frequency stretching deformation of piezoelectric elements. Such an approach effectively increases the power density of piezoelectric elements and reduces the design difficulty of following electric energy extraction circuits. Among these frequency up-converting oscillators, their internal resonance-based harvesters are specially introduced. To our knowledge, the mechanical FUC function based on internal resonance mechanism was firstly proposed by Wu et al. [52,53,54]. Comparing with the other FUC strategies based on mechanical impact and plucking mechanisms, noises and impacts are missing during the internal resonance based FUC process, and the required external acceleration threshold is much lower (0.05 g, as shown in Table 2). Besides, these harvesters with the pendulum mechanism are also very suitable for the ultra-low frequency (~2 Hz or even lower) vibrations energy harvesting; for instance, the proposed pendulum spring piezoelectric harvester could generate a high output power of 13.29 mW at conditions of 2.03 Hz and 0.26 g motion excitation, which show a great energy harvesting performance, and prospects in the power supply of autonomous devices in the field of Internet of Things.

## Figures and Tables

**Figure 1 micromachines-13-00210-f001:**
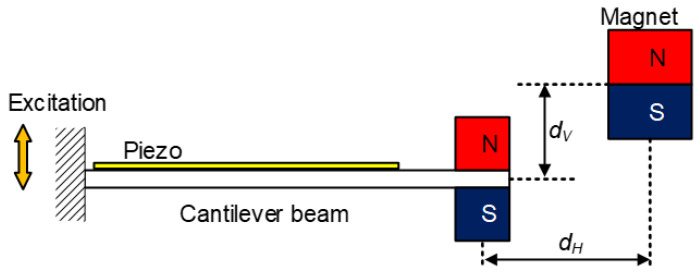
Resonant frequency tuning oscillator based on magnetic force.

**Figure 2 micromachines-13-00210-f002:**
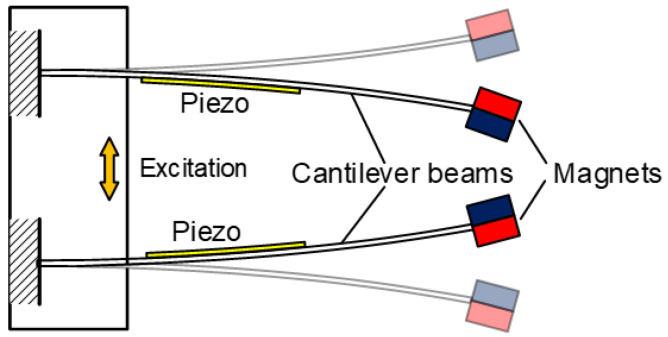
Resonant frequency tuning and multi-mode oscillator based on magnetic force.

**Figure 3 micromachines-13-00210-f003:**
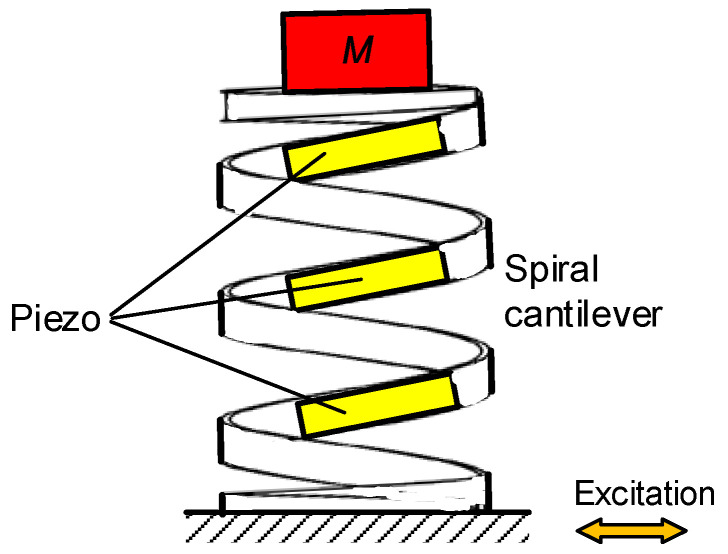
Multimodal piezoelectric oscillator based on a spiral cantilever structure.

**Figure 4 micromachines-13-00210-f004:**
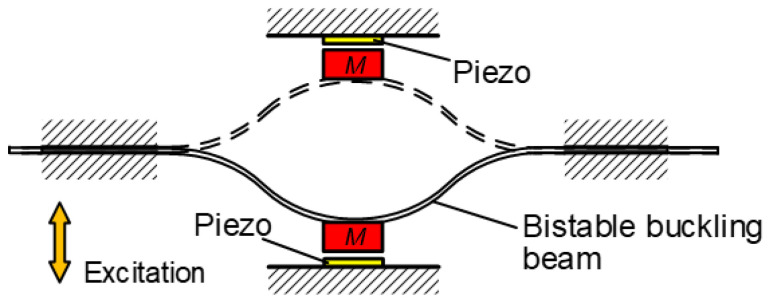
Nonlinear wideband oscillator realized by a purely elastic buckling beam.

**Figure 5 micromachines-13-00210-f005:**
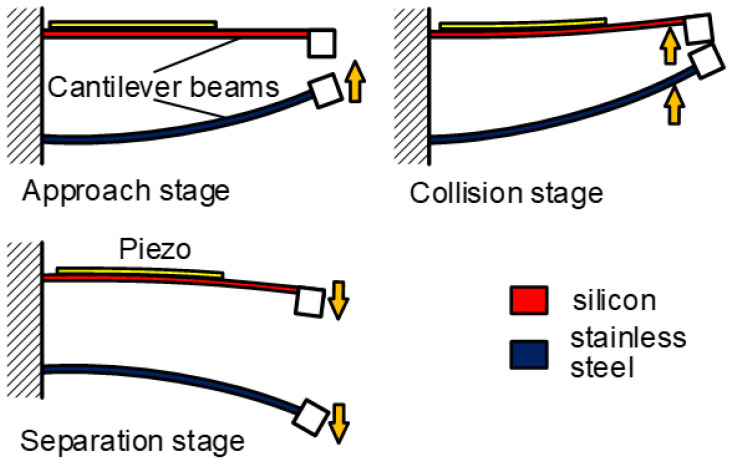
Frequency up-converting oscillator based on mechanical impact mechanism.

**Figure 6 micromachines-13-00210-f006:**
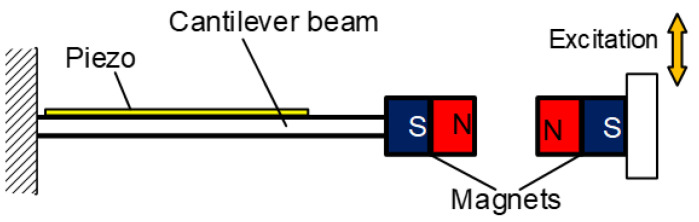
Frequency up-converting oscillator based on impulse-like magnetic force.

**Figure 7 micromachines-13-00210-f007:**
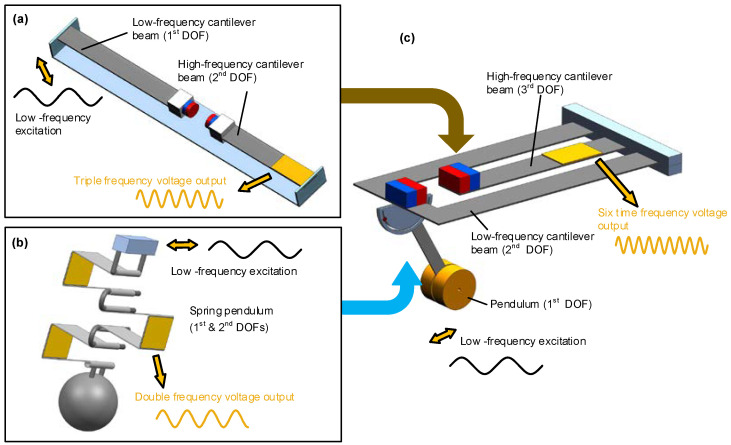
Frequency up-converting oscillators based on the internal resonance mechanism: (**a**) 1:3 internal resonance based oscillator; (**b**) 1:2 internal resonance based oscillator; (**c**) 1:2:6 internal resonance based oscillator.

**Figure 8 micromachines-13-00210-f008:**
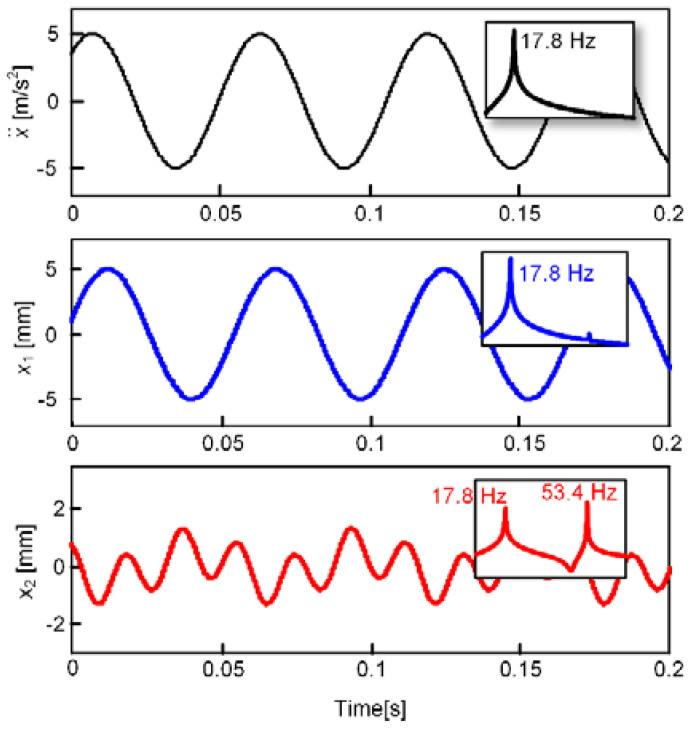
Waveforms and their frequency response curves under the sinusoidal excitation of which the acceleration amplitude is 5 m/s^2^.

**Figure 9 micromachines-13-00210-f009:**
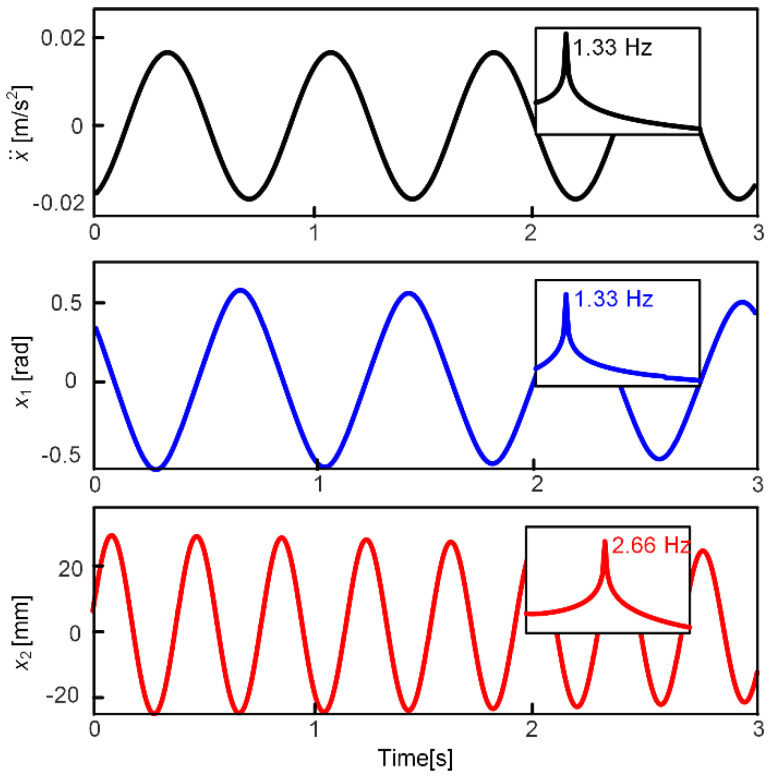
Waveforms and their frequency response curves under the horizontal excitation of which the frequency and acceleration amplitude are 1.33 Hz and 0.16 m/s^2^, respectively.

**Figure 10 micromachines-13-00210-f010:**
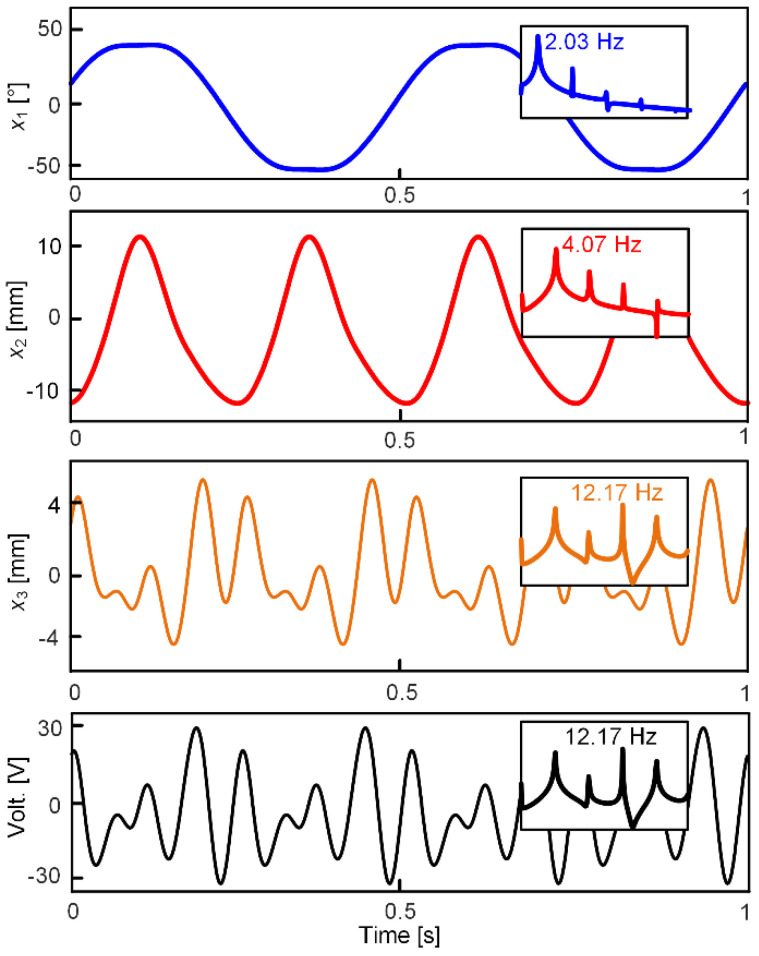
Waveforms and their frequency response curves under a horizontal excitation, of which the frequency and acceleration amplitude are 2.03 Hz and 2 m/s^2^, respectively.

**Figure 11 micromachines-13-00210-f011:**
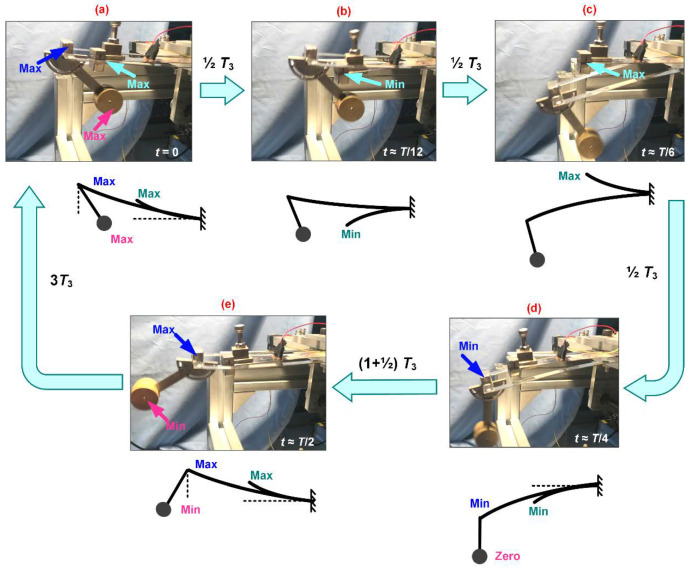
Experimental verification diagram of 1:2:6 internal resonance based frequency up-conversion.

**Table 1 micromachines-13-00210-t001:** Performance comparison of the oscillators based on resonant frequency matching strategy.

Ref.	Schematic Diagrams	Classification	Matching Mechanism	Operation Bandwidth
[22]	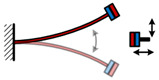	Resonant frequency tuning	Magnetic force induced stiffness variation	51~110 Hz
[23]	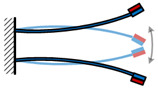	Resonant frequency tuning and multi-modal	Magnetic attractive force induced stiffness variation and multiple degrees of freedom	23~30 Hz
[28]	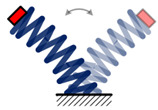	Multi-modal	Multiple degrees of freedom	Around 16, 21 and 28 Hz
[34]	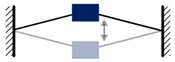	Nonlinear wideband	Mechanical load induced nonlinearity	4~12 Hz

**Table 2 micromachines-13-00210-t002:** Performance comparison of various FUC oscillators.

Ref.	Schematic Diagrams	Mechanism	FUC Factor	Excitation Source
[47]	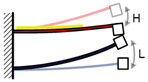	Mechanical impact	25.3 (40 Hz → 1012 Hz)	Acceleration excitation (0.3 g)
[50]	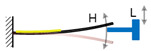	Mechanical plucking	~297 (~1 Hz → 297 Hz)	Human naturally walk
[51]	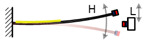	Impulse-like magnetic force	~2.5 (0.8 Hz →~2 Hz)	Excitation displacement (50 mm)
[44]	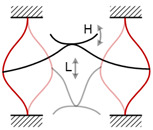	Impulse-like acceleration	6.4 (12 Hz → 77 Hz)	Acceleration excitation (0.5 g)
[52]	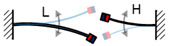	Internal resonance (1:3)	3.0 (7.78 Hz → 23.41 Hz)	Acceleration excitation (0.1 g)
[53]	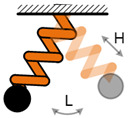	Internal resonance (1:2)	2.0 (1.83 Hz → 3.65 Hz)	Acceleration excitation (0.05 g)
[54]	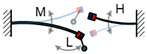	Internal resonance (1:2:6)	5.9 (2.00 Hz → 11.75 Hz)	Acceleration excitation (0.25 g)

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
