# Peer review of "Three Frequency Up-Converting Piezoelectric Energy Harvesters Caused by Internal Resonance Mechanism: A Narrative Review"

_micromachines, 2022, doi:10.3390/mi13020210_

Round 1
Reviewer 1 Report
This paper reviews the piezoelectric oscillator structures with frequency up-converting function, including three FUC piezoelectric oscillating structures , which were proposed by the authors' team. PLease make the following minor revisions to make the paper more readable.
- In the first column of Table 2, please indicate the low frequency vibrators , high frequency vibrators and the vibration directions of them.
- Please refine english of the text.
Author Response
Dear Reviewer,
Many thanks for your comments which are helpful for the quality of the manuscript.
We have modified the manuscript with all the changes marked in red. The point-by-point response is given in the corresponding attachment. Please check them.
We really appreciate your warm work earnestly and hope that the correction will meet with approval.
With our best regards,
Jian Liu and Yipeng Wu

Reviewer 2 Report
1) Please, inform the sofware usedd at simulations.
2) Why Do the authors only present simulation results (Fig. 8 to 10).
Author Response

(The authors gave the same response as above.)

Reviewer 3 Report
In this manuscript, the authors report a type of piezoelectric oscillator structures based on mechanical structures. It does not seem professional and logical. Additionally, there is nothing new. Therefore, my recommendation would be to reject this manuscript.
- Some English expressions are not concise enough, even wrong. For instance, “This paper mainly introduces a type of frequency up-converting piezoelectric energy harvesters based on an internal resonance mechanism, compared with the other frequency up-converting mechanisms caused by mechanical impact, mechanical plucking, the internal resonance based harvesters can up-convert the frequency under a condition of much lower excitation level and much less energy loss, wideband operation bandwidth which benefits to practical vibrations also exists in these multi-degree-of-freedom nonlinear dynamic systems.” ; “This paper summarizes a type of piezoelectric oscillator structures based on mechanical structures and these have a FUC function.”; “Then the paper focus on introducing three FUC piezoelectric oscillating structures caused by internal resonance mechanism in the multi-DOFs nonlinear system, which were proposed by our team members successively.”. The authors must look through this manuscript carefully and then make major revision.
- In the sentence “Moreover, the value of the frequency up-conversion factor based on internal resonance mechanism can reach as high as 6 so far, which is already a quite high value…”. How about the value of 6? Whether does it improve as compare with other references?
- Some subtitle is not standard, liking “0. Introduction”. It does not seem professional.
- In Section 2 “FUC oscillators based on internal resonance mechanism”, this paper has been devided into many subsections, such as “2.1 1:3 internal resonance system”, “2.2 1:2 internal resonance system”, and “2.3 1:2:6 internal resonance system”. What is the internal logic? Only the parameter variant? It easily makes the readers confused.
- Lastly, the authors claim that “Compared with traditional FUC strategies such as mechanical impact, mechanical plucking, and magnetic force plucking, the internal resonance FUC structure has advantages of lower excitation acceleration threshold, less energy loss, lower noise and better fatigue life.” Where are the detailed values, such as excitation acceleration threshold, less energy loss, lower noise and better fatigue life. Subsequently, how to quantify the differences between this type and the traditional FUC strategies?
Author Response

(The authors gave the same response as above.)

Reviewer 4 Report
Refer the attachment please.

Author Response

(The authors gave the same response as above.)

Reviewer 5 Report
This manuscript mainly talks about the frequency up-converting oscillators applied in low-frequency vibration energy harvesting. A special mechanism named internal resonancebased frequency up-conversion method was thoroughly discussed: the authors introduced the principle of the mechanical frequency up-conversion phenomena based on internal resonance and the three structure implementations and their energy harvesting performances. As the authors said, this mechanism has the advantages of lower excitation level, less energy loss, etc.
The manuscript was well written and interesting to read. However, there are a few comments or suggestions that should be addressed in the revised version.
(1)At the beginning of subsection 1.2, the authors said “the response amplitudes are usually large, leading to higher requirements of the structure fatigue life and the integration of the piezoelectric ceramic”. Do the authors think that the large piezoelectric strain will also easily induce the depolarization of piezoelectric ceramics?
(2)Following to the above question, please list some examples to explain the difficulty of improving the energy extraction efficiency.
(3)In Table 1, you mentioned the Refence 22, in which the operation bandwidth is 51-110 Hz. Compared with other similar researches validated by experiments, is this theorical result right? You should confirm this paper again.
(4)In Figure 10, it the piezoelectric output voltage an open circuit voltage or a voltage across the load resistor R?
(5)In page 12, first paragraph, there is no ‘Figure 13’ in the manuscript.
(6)Page 5, line 160, the author should confirm the content “from 1012 Hz to 40 Hz”. Is this statement right?
(7)In the conclusion part, the authors should point out the penitential applications such as in IOTs and the corresponding prospective in further research to let readers know the future direction of this research area. As this paper is a review.
Author Response

(The authors gave the same response as above.)

Round 2
Reviewer 4 Report
The authors have replied my all questions and revised the manuscript well. The manuscript could be accepted for publication.